# Foliar Application of Selenium Associated with a Multi-Nutrient Fertilizer in Soybean: Yield, Grain Quality, and Critical Se Threshold

**DOI:** 10.3390/plants12102028

**Published:** 2023-05-18

**Authors:** Maila Adriely Silva, Gustavo Ferreira de Sousa, Gustavo Avelar Zorgdrager Van Opbergen, Guilherme Gerrit Avelar Zorgdrager Van Opbergen, Ana Paula Branco Corguinha, Jean Michel Moura Bueno, Gustavo Brunetto, José Marcos Leite, Alcindo Aparecido dos Santos, Guilherme Lopes, Luiz Roberto Guimaraes Guilherme

**Affiliations:** 1Soil Science Department, Federal University of Lavras, Lavras 37200-900, Brazil; m.adriely@hotmail.com (M.A.S.); gustavoferreira_s@hotmail.com (G.F.d.S.); gustavo.opbergen1@estudante.ufla.br (G.A.Z.V.O.); guilherme.opebergen@estudante.ufla.br (G.G.A.Z.V.O.); anapaulacorguinha@hotmail.com (A.P.B.C.); guilherme.lopes@ufla.br (G.L.); 2Soil Science Department, Federal University of Santa Maria, Santa Maria 97105-900, Brazil; bueno.jean1@gmail.com (J.M.M.B.); brunetto.gustavo@gmail.com (G.B.); 3ICL South America, São Paulo 01310-200, Brazil; josemarcosleite@yahoo.com.br; 4Institute of Chemistry, University of São Paulo, São Paulo 05508-000, Brazil; alcindo@iq.usp.br

**Keywords:** selenate, biofortification, selenium reference values, cereal

## Abstract

Selenium uptake and its content in soybean grains are affected by Se application methods. This study evaluated the impact of Se foliar application combined with a multi-nutrient fertilizer (MNF) on soybean, establishing a Se threshold to better understand the relationship between Se content in grains and yield of two genotypes (58I60 Lança and M5917). Two trials were conducted in a 4 × 2 factorial design: four Se rates (0, 10, 40, 80 g Se ha^−1^) and two methods of foliar Se application (Se combined or not with MNF). Foliar fertilizers were applied twice, at phenological stages of beginning of pod development and grain filling. Grain yield increased with the application of MNF, yet Se rates increased Se contents linearly up to 80 g Se ha^−1^, regardless of the use of MNF. Lança and M5917 genotypes had grain Se critical thresholds of 1.0 and 3.0 mg kg^−1^, respectively. The application of Se favored higher contents of K, P, and S in grains of genotype Lança and higher contents of Mn and Fe in grains of genotype M5917. Our findings highlight the importance of addressing different Se fertilization strategies as well as genotypic variations when assessing the effects of Se on soybean yield and grain quality.

## 1. Introduction

Selenium (Se) is a trace element required by both humans and animals [1]. It acts as a co-factor for antioxidating enzymes (e.g., glutathione peroxidase) and has functions in immune system maintenance, cardiovascular disease reduction, thyroid regulation, detoxification capacity, and anti-cancer and anti-viral action [1,2]. The recommended daily dose of Se for adults is 60–70 μg day^−1^, and it is estimated that approximately one billion people worldwide are Se deficient [3,4]. Low Se intake in humans has been closely linked to Keshan and Kashin–Beck disease [5,6].

Food biofortification has been an alternative for reducing Se deficiency in the population. Biofortification is a process that increases the content of minerals and vitamins (such as Se, zinc—Zn, iodine—I, and iron—Fe) in edible parts of plants to improve nutritional quality for humans and animals [7,8,9]. Approximately 25 proteins of the human body have Se as a component [10]. It also aids in the regulation of thyroid hormone, DNA synthesis, and fighting free radicals [10,11]. Selenium deficiency in the body has been linked to several diseases, including neurological disorders, cancer, and heart disease [5,12,13].

Selenium is not considered a nutrient for plants, but it has beneficial effects, such as reducing the stress caused by low temperatures in coffee [14], increasing the yield of wheat [15], increasing protein and amino acid content in soybean [16], and reducing drought stress in common bean plants [17]. Its effects on plants have been studied for approximately 70 years, and several advantages have already been reported [14,18,19,20,21]. Due to its beneficial effects on plants, recently a change of the definition of “plant nutrient” has been proposed to start a debate on the inclusion of Se, and other beneficial elements, as a plant nutrient [22].

Selenium bioavailability and transfer into the food chain are influenced by soil biophysical chemistry, presence of competing ions, plant species/genotype, method of application, and Se rates applied [9,21,23,24]. Tropical soils have a high capacity for retaining Se—due to the presence of Fe/Al oxyhydroxides—[25,26], as a result, foliar spray application has been considered as one strategy to overcome Se sorption in soils and to increase the efficiency of biofortification practices [24,27,28].

In contrast to soil application, foliar spray improves Se uptake and recovery efficiency by reducing Se immobilization in the soil and shortening the transport distance of Se from plant roots to shoots [29]. In addition, Se spray associated with a multi-nutrient fertilizer may be a promising strategy for increasing Se uptake/redistribution by plants. However, it is critical to understand the Se content in plant tissue to ensure that it does not reduce yield by causing toxicity. High Se concentrations in plants can cause phytotoxicity by directly affecting metabolism, resulting in chlorosis, cell membrane degradation, senescence, and reduced growth and grain yield [30]. Additionally, in the current Brazilian recommendation systems, there are no reference values (critical threshold) of Se for the soybean crop. Furthermore, it is unclear if there are critical threshold variations among soybean genotypes. These gaps can be filled by Bayesian modeling techniques combined with well-documented databases [31,32]. 

Soybean (*Glycine max* [L. Merril]) is the most widely grown crop in the world, and its grains are high in protein (about 40 percent) [33]. Soybean grains are used to produce feed for animals, in order to produce meat for human consumption. Furthermore, soybean plays a variety of roles in food processing due to its distinct protein-related food texture, high water-holding capacities, and foaming properties [34,35]. As a result, the current study aimed to assess the impact of Se foliar application associated with the use of a multi-nutrient fertilizer on soybean yield and grain quality. In addition, we want to establish a grain-Se critical threshold to understand the relationship between Se content in soybean grains and the yield of different soybean genotypes.

## 2. Results

### 2.1. Variance Components

The results obtained showed a great influence of the genotypes for most of the analyses performed (Figure 1). They were supported by principal component analyses, in which the results were also separated by genotype (Appendix A). On the other hand, Se content in grains was affected mainly by Se rates.

### 2.2. Grain Yield

The spraying of Se rates did not affect grain yield (*p* > 0.05). On the other hand, MNF application influenced the yield of both genotypes (*p* < 0.05) (Figure 2A,B). The application of Se and MNF increased ~0.32 and 0.38 t ha^−1^ of grains (i.e., ~5.37 and 6.26 60-kg bags ha^−1^) for 58I60 Lança and M5917 genotypes, respectively, compared with the application of Se without MNF. In general, the genotype M5917 showed the highest grain yield—5.0 t ha^−1^ on average—while the genotype 58I60 Lança produced 4.2 t ha^−1^. Average yields with the MNF application varied from 4.3 to 4.7 t ha^−1^ for genotype 58I60 Lança, and from 5.0 to 5.4 t ha^−1^ for genotype M5917.

### 2.3. Selenium Content and Se Recovery in Grains

Selenium content in grains was affected by the interaction between Se rates and MNF application (Figure 3A,B). In both genotypes, increasing Se rates increased the Se content linearly up to the highest rate (80 g Se ha^−1^), regardless of the use of MNF. In 58I60 Lança, the increase in grain Se content for each gram of Se applied via foliar was 0.063 mg kg^−1^ with MNF and 0.055 mg kg^−1^ without MNF. In genotype M5917, the Se content in grain increased by 0.087 mg kg^−1^ with MNF and 0.065 mg kg^−1^ without MNF for each gram of Se applied through foliar supply. According to the confidence interval (95%), the application of MNF did not affect the Se content in the soybean grains up to the rate of 43.8 g Se ha^−1^ for genotype 58I60 Lança and 29.4 g Se ha^−1^ for genotype M5917. However, after these Se rates, the association of Se with MNF promoted higher Se content.

The Se recovery rate (%) by soybean grains was also affected by Se foliar rates (*p* < 0.05) (Figure 3C,D). The Se recovery in genotype 58I60 Lança at the rate of 10 g Se ha^−1^ was 18.4%, whereas the application of 80 g Se ha^−1^ promoted 24.1% of Se recovery. The foliar rates of 10 and 80 g Se ha^−1^ showed different Se incorporation by the grains, but both were statistically equal to the foliar rate of 40 g Se ha^−1^. For genotype M5917, the Se foliar rates of 40 g Se ha^−1^ and 80 g Se ha^−1^ promoted equal Se rate recovery (*p* > 0.05), 31.1% and 37.7%, respectively. The Se recovery obtained with the application of Se foliar at 10 g Se ha^−1^ was 21.5%.

### 2.4. Macronutrients, Micronutrients, Proteins, and Amino Acids

The content of K, P, and S in the grains was affected by the interaction among the studied factors (Se foliar rates and MNF) in the 58I60 Lança genotype (*p* < 0.05) (Appendix A). Despite the significant difference in the K and P content in the soybean grains, the data did not fit either the linear or the quadratic regression model. In addition, the content of K and P were higher in the grains that had received Se foliar at 80 g Se ha^−1^ combined with the MNF. 

For genotype M5917, the F test for Cu and Zn showed significant differences among Se foliar rates and MNF (Appendix A). On the other hand, the content of K, Mn, P, S, and N were affected by the Se rates (*p* < 0.05). The content of Mg, Ca, and Fe was not affected by the factors studied (*p* > 0.05). Following in the genotype M5917, the K content in the grains fitted quadratic regression in the levels of Se supplied via foliar application (R^2^ = 77%). The highest K content in grains was 18.57 g Se ha^−1^ and it was obtained by the application of 45.6 g Se ha^−1^. 

Concerning genotype 58I60 Lança, the S content increased linearly upon increasing the Se rate, regardless of the MNF application, with a correlation coefficient of 89% being observed with MNF and 68% without MNF. The MNF promoted a higher accumulation of S in grains, regardless of the Se rate in genotype M5917. The S data were fitted to a quadratic model (R^2^ = 63%), and the highest S content was obtained by the rate of 37 g Se ha^−1^.

In M5917, the interaction between Se rates and MNF affected the Zn and Cu content in grains. Additionally, N and Mn contents were affected by the Se rates, with Se rates providing a quadratic regression fit for N content in grains. Se foliar and MNF application did not influence the N, Cu, and Zn in the 58I60 Lança. Protein contents in soybean grains increased quadratically upon increasing Se rates in the M5917 (*p* > 0.05) (Appendix A). The total free amino acids content was affected by the interactions among Se rates and MNF, yet there was a fitted model regression only in genotype 58I60 Lança.

### 2.5. Selenium Critical Threshold and Selenium Intake (µg person^−1^ day^−1^)

The critical Se threshold in soybean grains was estimated at 1.0 mg kg^−1^ and 3.0 mg kg^−1^ for genotypes Lança and M5917, respectively (Figure 4A,B). This means that above 1.0 mg kg^−1^ of Se in the grain the yield in the genotype 58I60 Lança is reduced, whereas the same effect occurs for genotype M5917 above 3.0 mg kg^−1^. Therefore, genotype M5917 is more tolerant to Se accumulation in grains than 58I60 Lança.

Figure 4C,D show the relationship between Se rate and human daily Se intake based on an average recommended intake of soybean protein (25 g person^−1^ day^−1^). As per the estimated daily Se intakes shown in Figure 4C,D, the rate of Se that should be supplied to the plant in order to obtain a Se content in soybean grains suitable for human consumption (considering a recommended daily intake of 70 μ of Se day^−1^) would be: 15.1 g Se ha^−1^ (with MNF—genotype 58I60 Lança), 16.2 g Se ha^−1^ (without MNF—genotype 58I60 Lança), 15.3 g Se ha^−1^ (with MNF—genotype M5917), and 15.4 g Se ha^−1^ (without MNF—genotype M5917).

Considering the daily Se consumption by person, the adequate Se rates to produce enriched soybean grains are below the Se content that reduces crop yield (below the Se critical threshold). Indeed, using the model fitted for Se content in grains (Figure 2A,B), Se rates to promote 1 mg kg^−1^ in grains were 19.5 g Se ha^−1^ (with MNF) and 23.3 g Se ha^−1^ (without MNF) for the Lança. In the genotype M5917, the rates to produce soybean grains with 3 mg kg^−1^ of Se are 36.1 g Se ha^−1^ (with MNF) and 49.3 g Se ha^−1^ (without MNF).

### 2.6. Pearson’s Correlation Matrix

The correlation between the variables assessed is presented in Figure 5. In genotype Lança, Se content correlated positively with S (R^2^ = 65%), P (R^2^ = 57%), and K (R^2^ = 57%) content in grains. For genotype M5917, the increase of Se in the grains increased Mn content (R^2^ = 44%), and Fe (R^2^ = 51%). Selenium content in grains negatively affected yield and total free amino acid content in the genotype Lança (R^2^ = 29% and 73%, respectively). This effect was not observed for cultivar M5917.

## 3. Discussion

Selenium is not considered a nutrient for plants, yet its beneficial effects are already well established [22,36]. The increase of the content of a target element (e.g., Se) in the edible part of the plant without reducing yield is one of the key assumptions for the implementation of biofortification programs. Our results showed significant differences in yield only due to the application of MNF (Figure 2A,B). Although there was no difference with Se foliar spray in soybean, some studies showed increased yields of wheat [15,37], rice [19], and coffee [38]. Such increases are mostly observed in plants under biotic and abiotic stress conditions [17,39,40]; discrepancies in results may be attributed to the various growth stages and methods of Se application in plants.

Foliar spraying with Se resulted in a linear increase of the Se content in soybean grains (Figure 3A,B). Similar results were observed in wheat [15] and rice [24]. The lowest rate at which it is possible to see a difference in Se content among the treatments (with MNF and without MNF) is 43.8 kg Se ha^−1^ for Lança and 29.4 kg Se ha^−1^ for genotype M5917, which is about 33% less, indicating that the application is more efficient for the last genotype. This also demonstrates that the interaction of Se foliar and other nutrients can enhance the uptake/redistribution of Se in plants. 

Nitrogen is one of the components of MNF and it is known that it can affect Se uptake by plants [41]. The fact that Se and S use the same metabolic pathway in plants can be used to explain this interaction between N and Se. Applications of N enhance O-acetyl serine, an important regulator of S metabolism in cysteine synthesis in plants, which then increases the synthesis of cysteine and protein. Along with N, the MNF also contains Mg, K, and B, which are important nutrients for the transport of photoassimilates within plants [42,43,44]. Magnesium has a direct impact on the yield and quality of grains. A lack of Mg both affects the transport of assimilates from the leaves to the grains and reduces the transport of amino acids in plants [45].

Previous studies have produced conflicting findings regarding the response of various essential elements to Se treatment in several crops. In rice, Se application combined with N fertilization resulted in higher grain concentrations of N, P, K, Ca, Se, B, Cu, Zn, and Mo [41]. Wheat cultivated under drought stress showed increased Fe content and decreased Zn content on shoot with Se foliar spray [46]. The presence of Se in plants can alter the ionic permeability coefficient in the plasma membrane, thereby altering the transport and accumulation of micronutrients in plant cells [47]. Nonetheless, the mechanisms by which Se interferes with other elements require further investigation.

Several factors influence selenium recovery, including the Se source and the plant species. In a previous study carried out by our research group, the recovery of Se applied via soil was 6.5% for genotype M5917 and 8.1% for genotype Lança [21]. In the current study, the average recovery rate of Se applied via foliar was 24.1% for Lança and 37.7% for M5917 (3–5 times higher), which was remarkably higher than the wheat’s maximum recovery of Se by grains (3%) [15]. 

Changes in the content of the nutrients in grains are shown in Appendix A, suggesting potential interactions between Se and these nutrients. In fact, providing Se to plants may affect the foliar absorption of selected nutrients or their redistribution within plants, which in turn affects the concentration of those nutrients in soybean grains. Djanaguiraman et al. (2010) [48] reported that the availability of Se to plants can have an impact on the uptake and accumulation of the nutrients necessary for plant metabolism. The interaction between Se and S is the most studied, due to the similarity between these elements; they share the same metabolic pathway in plants [49,50]. In general, Se foliar supply affected the content of P and K in grains of the genotype Lança, as well as N, P, K, S, Cu, Mn, and Zn in grains of the genotype M5917. Except for Cu, Mn, and Zn, the other nutrients affected by Se application are those found in MNF. 

The critical Se threshold demonstrated that above 1.0 mg kg^−1^ of Se in grain, the yield in the genotype 58I60 Lança is reduced, while genotype M5917 showed this effect only above 3.0 mg Se kg^−1^, which is a clear indication that genotype M5917 is more tolerant to Se accumulation in grains than 58I60 Lança. Selenium toxicity in plants has not been fully investigated, but it has been demonstrated that tolerance varies depending on plant species and genotypes [30]. Indeed, a previous experiment with two soybean genotypes demonstrated such variation in Se response [51], with the Sonja genotype being more sensitive to Se and displaying a stronger physiological response when compared with the Lucija genotype.

In another study assessing Se toxicity to various crops, brown mustard has shown a greater tolerance to Se when compared with maize, rice, and wheat [52]. From the established critical level of Se by the authors, there was a reduction of 21% in the dry matter of brown mustard, 24% in maize, 11% in rice, and 27% in wheat. These levels where related to the following Se content in the shoot dry matter of the studied crops: 18.9 mg kg^−1^ for wheat, 41.5 mg kg^−1^ for rice, 76.9 mg kg^−1^ for corn, and 104.8 mg kg^−1^ for brown mustard [52].

As seen in our study, Se intake values for biofortifying soybean, considering the selected data for adequate daily intake and average consumption of soy protein, are lower than the level of Se in grains that reduced yields (Figure 4C,D). This fact is relevant since it is possible to biofortify soybeans without decreasing the grain yield of the crop. The dietary habits of the population have a significant impact on Se intake by the population. Excessive human Se intake, typically greater than 400 μg day^−1^, may lead to toxicity, resulting in health problems. Some of the Se toxicity symptoms are hair and nail loss, skin lesions, nervous system dysfunction, and even death [53,54]. As estimated by the regression equations shown in Figure 4C,D, our findings indicate that lower Se rates can satisfy smaller demands. However, it must be noted that these estimates assume that soybean will be consumed by humans as soybean protein and do not account for possible Se losses during other industrial processes.

## 4. Materials and Methods

### 4.1. Growth Conditions and Experimental Design

Two similar trials were conducted on a soybean field during the 2018/2019 cropping season, at Farm Uva, municipality of Capão Bonito, São Paulo state, Brazil (Lat: −24.040934, Lon: −48.262421), with an average annual rainfall of 1628 mm and an average annual temperature of 18.8 ºC, under a humid subtropical climate (Cfa) [55]. The soil is classified as Oxisol (Typic Hapludox) [56] and its chemical and physical characteristics are as follows [57]: pH (H_2_O) = 6.0; H + Al = 2.96; Al = 0.06; P (Mehlich-1) = 34.8 mg dm^−3^; K = 148 mg dm^−3^; S = 4.11 mg L^−1^; CEC = 9.83 cmol_c_ dm^−3^; Ca = 5.05 cmol_c_ dm^−3^; Mg = 1.44 cmol_c_ dm^−3^; P-rem = 28.10 mg L^−1^; organic matter = 2.69 dag dm^−3^; clay = 510 g kg^−1^; silt =110 g kg^−1^; and sand =380 g kg^−1^.

Each experiment was carried out with one soybean genotype, 58I60 Lança, and M5917. The experiment was arranged in a randomized block with a full factorial design of 4 × 2, being four Se rates (0, 10, 40, and 80 g Se ha^−1^) and two methods of Se application: (i) Se with multi-nutrient fertilizer (hereafter called MNF) and (ii) Se without MNF. The experiments were composed of four blocks, totalizing 32 experimental plots for each genotype. The treatments with MNF received 2 kg ha^−1^ of the product; its composition was as follows: nitrogen (N—5%), phosphorus pentoxide (P_2_O_5_—10%), potassium oxide (K_2_O—20%), magnesium (Mg—29%), sulfur (S—12%), and boron (B—0.5%). Sodium selenate was used to prepare the Se solution (Na_2_SeO_4_ Sigma-Aldrich, Saint Louis, MO, USA). Foliar spray fertilizers were applied twice, at phenological stage R3 (beginning of pod development) and R5 (grain filling), with each application containing half of the total dose. Sodium selenate and MNF (if used) were diluted in deionized water, mineral oil was added, and a pressured carbon dioxide pump was used to apply it.

The plots were composed of 4 sowing lines with a spacing of 0.5 m between lines and 7 m in length. The collection of material for analysis was carried out in the useful area of the plot. For the composition of the useful area of the plot, 0.5 m from each end of the plot and two lateral lines were disregarded. Fourteen seeds were sown per meter and fertilization was carried out with 16 kg ha^−1^ N, 80 kg ha^−1^ P_2_O_5_, and 28 kg ha^−1^ K_2_O.

### 4.2. Yield

The grains from the useful area of each experimental plot were harvested after maturity and soybean yield was calculated by weighing the grains and extrapolating to a hectare (kg ha^−1^), considering 13% moisture. After that, the grains were dried until they reached a constant weight and were ground with an electric hand mill. 

### 4.3. Selenium, Macronutrient, and Micronutrient in Soybean Grains

To determine Se, macronutrients, and micronutrients, 0.5 g of samples were placed in Teflon vessels and 5 mL of HNO_3_ (65%) was added. The extract was allowed to stand at room temperature overnight before being digested the next morning (16 h). The vessels were then hermetically sealed and heated in a Mars-5 microwave digestion oven (CEM Corp, Matthews, NC, USA) following the 3051A methodology, proposed by the United States Environmental Protection Agency [58]. The content of macronutrients and micronutrients was obtained using inductively coupled plasma—optical emission spectroscopy (ICP-OES). 

Selenium content in the solutions was measured using a graphite furnace atomic absorption spectrometer (Atomic Absorption Spectrometry with Zeeman background correction and EDL lamp for Se; AAnalyst™ 800 AAS, Perkin Elmer, Waltham, United States). The calibration curve for Se measurement was obtained from a standard solution containing 1 g kg^−1^ of pure Se (Fluka, Buchs, Switzerland). To maintain digestion quality, each batch of digestion included standard reference material from the Institute for Reference Materials and Measurements (White Clover—BCR 402, IRMM, Geel, Belgium) and a blank sample. The main recovery value for standard material was 95% (*n* = 5). The detection limit (LOD) was calculated by taking the standard deviation and mean of 7 blank extracts. 

The fraction of the applied Se incorporated in soybean grains (Se recovery) was calculated using Equation (1) described below:(1)$$\mathrm{Se}\;\mathrm{recovery}\;\left(\%\right)=\frac{\left(\mathrm{Se}\;\mathrm{treatment}\;-\;\mathrm{Se}\;\mathrm{control}\right)}{\mathrm{Se}\;\mathrm{rate}\;}\;\times \;100$$
where: Se recovery (%) = use efficiency of the Se rates applied in the leaves by soybean grains (Se utilization percentage); 

Se treatment (g Se ha^−1^) = Se content in soybean grains from soybean plants grown in treatments that had received Se applications, considering the yield obtained in each treatment; 

Se control (g Se ha^−1^) = Se content in soybean grains from soybean plants grown in treatments without Se applications, considering the yield obtained in each treatment;
(2)Se intake=(100 × 25/ Prot) × Se
where: Se intake (µg person^−1^ day^−1^) = estimated daily Se intake per person; 

Se (µg kg^−1^) = the amount of selenium in soybean grains; 

Prot (g) = the average amount of protein in soybean grains (42.4 to Lança and 41.8 to M5917). 

### 4.4. Free Total Amino Acids and Protein

The ninhydrin method was used to determine total free amino acids [59]. The protein content of the grains was calculated by multiplying the N content by 6.25.

### 4.5. Statistical Analysis

The obtained data were analyzed for normality (Shapiro–Wilk test) and variance homogeneity (Bartlett’s test). They were then analyzed using analysis of variance (ANOVA), and when significant, linear, or quadratic regression models were fitted. The linear models were compared to Se content in grains, using a confidence interval, created at 95% of probability. The treatments were compared using the Tukey test (*p* < 0.05) for grain yield and Se recovery analysis. Pearson’s linear correlation matrix (*p* < 0.05) was also used to validate clusters and potential relationships of Se rates applied and plant attributes. The R software was used to carry out the analyses [37]. Subsequently, the obtained data were submitted and an analysis of variance components was performed to quantify the percentage contribution of each source of variation (genotypes, MNF, Se rates, block, interactions, and residues) on the total variance of each response variable (yield, protein, amino acids, and nutrients concentration). This statistical procedure was performed using the VCA package of the R statistical environment [60].

In this study, we used Bayesian models, which allow us to explore all the possible regression lines (combinations of intercepts, slopes, and breakpoints). The yield of soybean present in the database was converted into relative yield (%) by taking into consideration each genotype and crop season to develop critical threshold estimation models. Models were developed on the boundary line [61,62] using Bayesian segmented quantile regression [31] to measure the association between dependent variables (yield) and Se concentration in soybean grains. Bayesian analysis was used to adjust the parameters of the regression models [63]. In this adjustment, the Monte Carlo simulation with Markov chains (MCMC) [64] was used based on the Gibbs sampling algorithm, with 20,000 random drawings after a warm-up period of 10,000 iterations. The sampling stage was performed through normal distribution, based on the distribution a posteriori of Se concentration. Modeling was implemented by using the ‘rjags’ package [65] in the R software (R version 4.2.0) [60]. Critical levels were assumed as the point at which the adjusted line reached the plateau and did not show a further increase in crop yield, with the increase in nutrient concentration. Finally, density frequency was analyzed, at a 90% confidence interval, to determine Se borderline concentrations and highest density.

## 5. Conclusions

Se supply through foliar application in soybean plants is a notably good strategy to improve intake of Se by the population, and an association with other nutrients can improve the efficiency of biofortification strategies. Despite this, Se foliar applications should be used cautiously to avoid Se toxicity and yield loss in plants. The grain yield was higher with MNF application. In both genotypes, the Se rates increased the Se content linearly up to the highest rate (80 g Se ha^−1^), regardless of the use of MNF. The Lança and M5917 genotypes of soybean grains had Se critical thresholds up to 1.0 mg kg^−1^ and 3.0 mg kg^−1^, respectively. The application of Se increased the content of K, P, and S in the grains of the genotype 58I60 Lança as well as the content of Mn and Fe in the grains of the genotype M5917. Finally, we recommended applying 15.1 g Se ha^−1^ to genotype 58I60 Lança and 15.3 g Se ha^−1^ to genotype M5917, combined with MNF to improve the Se content in the grains. 

## Figures and Tables

**Figure 1 plants-12-02028-f001:**
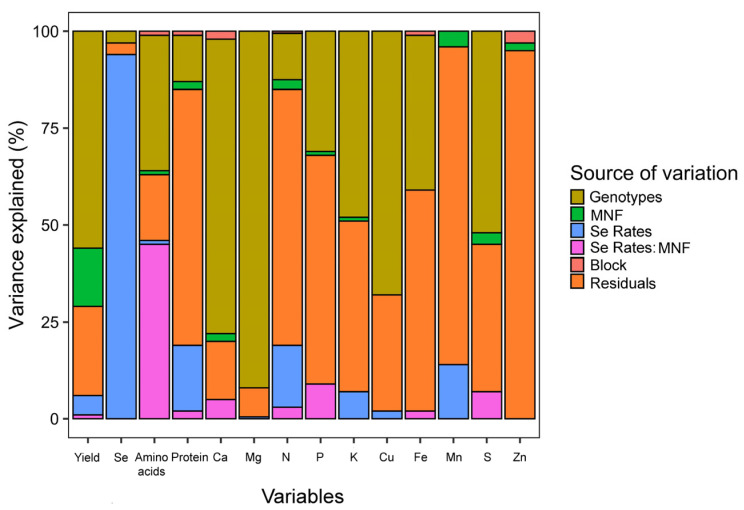
Visual representation of variance components. The colors represent the source of variation (genotypes, MNF, Se rates, block, and residuals). The variation proportion explained by each source of variation for each response variable is observed on the Y-axis (percentage).

**Figure 2 plants-12-02028-f002:**
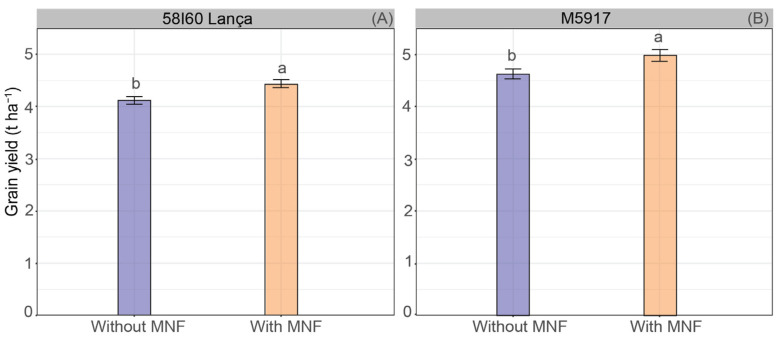
Grain yield (t ha^−1^) of soybean plants of genotypes 58I60 Lança (**A**) and M5917 (**B**). Lowercase letters compare the application of Se associated or not with MNF, at the 5% significance level, according to Tukey’s test. Vertical bars refer to the standard error (*n* = 16).

**Figure 3 plants-12-02028-f003:**
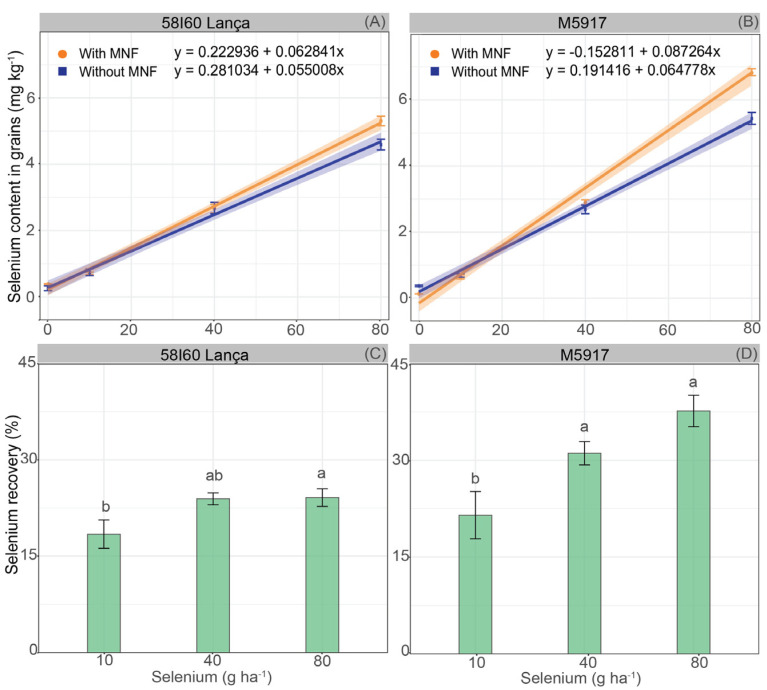
Selenium content in grains (mg kg ^−1^) (**A**,**B**) and Se recovery by soybean grains (%) (**C**,**D**) of soybean plants of genotypes 58I60 Lança and M5917. Lowercase letters compare Se rate, at the 5% significance level, according to Tukey’s test. Vertical bars refer to the standard error (*n* = 4 for selenium content in grains and *n* = 8 for selenium recovery).

**Figure 4 plants-12-02028-f004:**
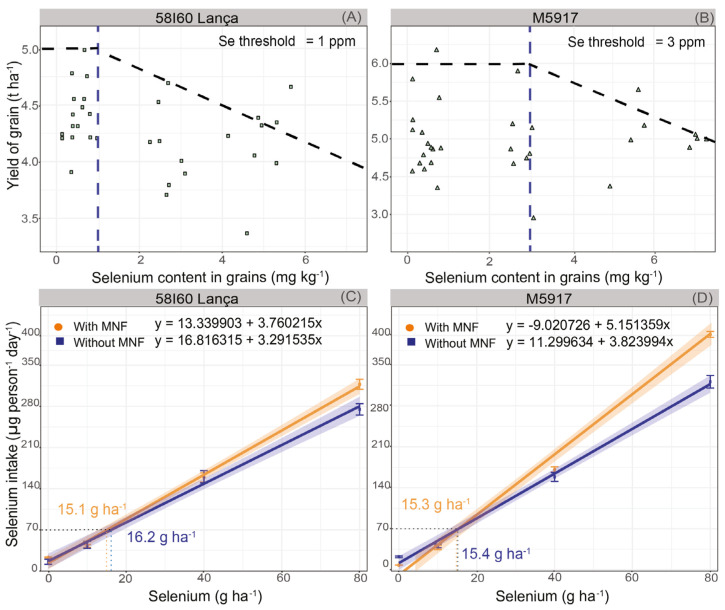
Selenium critical threshold in soybean grains (**A**,**B**) and selenium intake by a person by day (**C**,**D**) in the genotypes 58I60 Lança and M5917. Vertical bars refer to the standard error (*n* = 4). Triangles and boxes refer to grain yield values. The Blue dotted line refers to the Se critical threshold in the grain.

**Figure 5 plants-12-02028-f005:**
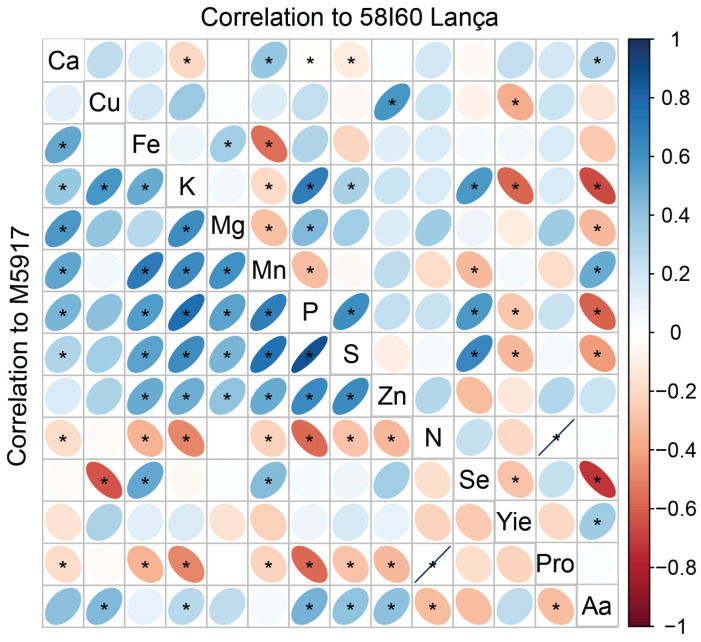
Pearson’s correlation matrix among the variables at 58I60 Lança and M5917 genotypes. Yie = yield; Pro = protein; Aa = amino acids. * means a positive Pearson’s correlation.

## Data Availability

The data presented in this study are available from the corresponding author upon reasonable request.

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
