# Peer review of "Foliar Application of Selenium Associated with a Multi-Nutrient Fertilizer in Soybean: Yield, Grain Quality, and Critical Se Threshold"

_plants, 2023, doi:10.3390/plants12102028_

Round 1
Reviewer 1 Report
The manuscript presents the results of an interestingly designed experiment to evaluate the effect of Se foliar application associated with the use of a a multi-nutrient fertilizer on soybean yield and grain quality. In addition, the authors in this study in order to understand the relationship between Se content in soybeans grains and yields of different soybean genotypes, established a critical threshold of Se in grain.
I find this manuscript valuable and interesting, due to the fact that the results of the experiment may be useful for improving the nutritional value, functionality and health-promoting perspectives of soybean seeds. Nevertheless, I have a few comments on it.
In the Introduction, please also write about the toxicity of Se to different crops, as the manuscript also contains the results of the critical Se threshold in soybean grains.
Line 47-49. I suggest rewording this sentence so that the quotation is limited to square brackets only, as in the rest of the manuscript. For example: Due to its beneficial effects on plants, it has recently been proposed to change the definition of "plant nutrient" and start a debate on the inclusion of Se - and other beneficial elements - as a plant nutrient [18].
In the Materials and Methods in Section 4.1. Growth conditions and experimental design, please clearly state in how many replicates the experiment was conducted and in total in how many experimental plots.
Line 103-104. ... 0.055 mg 103 kg-1 with MNF and 0.063 mg kg-1 without MNF - check for consistency with Fig. 3A
Line 105. ... 0.065 mg kg-1 with MNF and 0.087 mg kg-1 without MNF - check for consistency with Fig. 3B
Line 126. Instead of: The content of K, P, e S ..., should be: The content of K, P, and S
Line 202. Instead of: ... 43.8 kg Se ha-1 for Lança and 29.4 kg Se ha-1, ..., I propose: ... 43.8 kg Se ha-1 for Lança and 29.4 kg Se ha-1for cultivar M5917
Author Response
Dear Reviewer,
Thank you for your suggestions. It is attached our comments and corrections.

Reviewer 2 Report
Se will be toxic for human and plants, it should be careful to recommend to apply into crops due to the research do not give evidence that Se is insufficient in soil
1. Line 16: The background here did not show a sufficient need for this study. This should be rewritten or another supporting statement should be added.
2. The source of the two soybean genotypes should be briefly mentioned in the abstract.
3. Lines 39-49, this part also needs to be rewritten. At first glance, you give food biofortification to increase Se content in foods for human, but then you discuss its benefits for plants, so they are not supporting to each other. This can reduce the readability of your paper. You must choose between benefits for plants or human-beings in a paragraph, instead of merging them in the same paragraph, then another idea in another one.
4. What retains the Se in soil? It should be clarified.
5. About the threshold, what are the consequences if the Se application is excessive?
6. Lines 86-88, the increase was compared to what?
7. In Figure 1, Se content in grain was affected by genotypes, residuals, and Se rates, but in lines 100-101, the grain Se was affected by both Se rates and MNF? This difference can reduce the reliability of your result. It should be explained well.
8. In lines 158-162, the recommended intake of protein and Se should be cited.
9. In lines 225-227, you compare the two soybean cultivars, then compare the wheat’s Se recovery rate. In my opinion, this is inappropriate. I suggest to compared with another soybean cultivar or “In the current study, the average recovery rate of Se applied via foliar was 24.1%-37.7% (3-5 times higher) which was remarkably higher than the wheat’s maximum recovery of Se by grains (3%) [11].”.
10. In lines 281-285, on what basis do you choose the four different Se rates? Moreover, what is the literature for the MNF formula?
11. Lines 258-267, these statements need their own citations.
Author Response

(The authors gave the same response as above.)

Reviewer 3 Report
Silva et al. studied the effect that Se application methods and soybean genotypes has on Se uptake and Se grain content. Authors concluded by underlying the relevance to set up different Se fertilization schemes (by also taking into account genotypes differences) when assessing the effect of Se impact on grain yield and quality.
The rationale for study is clear, the title clearly reflects the content and the abstract is sufficiently informative, the introduction has appropriate background and the objectives are clearly described. The methodology of this study is routine, but it appears to be carefully carried out. On the whole the work in this manuscript is clearly presented and logical as the authors have a consolidate experience in this field. It appears to be carefully carried out, and I was unable to detect any significant deficiency.
Author Response
Dear Reviewer,
Thank you for your suggestions.
Round 2
Reviewer 2 Report
In the revised version, I found that MS has been significantly improved by the authors.